# Unraveling the Antihyperglycemic Effects of Dipeptyl Peptidase-4 Inhibitors in Rodents: A Multi-Faceted Approach Combining Effects on Glucose Homeostasis, Molecular Docking, and ADMET Profiling

**DOI:** 10.3390/ph18101589

**Published:** 2025-10-21

**Authors:** Raquel N. S. Roriz, Claudia J. P. Cardozo, Gabriela A. Freire, Caio B. R. Martins, Raimundo Rigoberto B. X. Filho, Landerson Lopes Pereira, Gisele F. P. Rangel, Tiago L. Sampaio, Lyanna R. Ribeiro, Gisele Silvestre Silva, Isabelle Maia, Deysi Viviana Tenazoa Wong, Daniele O. B. Sousa, Ariclécio Cunha de Oliveira, Eduardo Reina, Lidia Moreira Lima, Walter Peláez, Matheus Nunes da Rocha, Márcia Machado Marinho, Hélcio Silva dos Santos, Emmanuel Silva Marinho, Jane Eire Silva Alencar de Menezes, Fátima Regina Mena Barreto Silva, Kirley Marques Canuto, Nylane M. N. Alencar, Marisa Jadna Silva Frederico

**Affiliations:** 1Laboratório de Farmacologia Bioquímica, Departamento de Fisiologia e Farmacologia, Núcleo de Pesquisa e Desenvolvimento de Medicamentos (NPDM), Faculdade de Medicina, Universidade Federal do Ceará, Fortaleza 60430-275, CE, Brazil; raqroriz@gmail.com (R.N.S.R.); clajope19@gmail.com (C.J.P.C.); gabrielaafreire@alu.ufc.br (G.A.F.); landersonplopes@gmail.com (L.L.P.); giseleufc27@gmail.com (G.F.P.R.); gihchemistry@gmail.com (G.S.S.); daniele.sousa@ufc.br (D.O.B.S.); nylane@ufc.br (N.M.N.A.); 2Programa de Pós-Graduação em Ciências Naturais, Universidade Estadual do Ceará, Itaperi, Fortaleza 60714-903, CE, Brazil; rigobertoembrapams@gmail.com (R.R.B.X.F.); nunes.rocha@aluno.uece.br (M.N.d.R.); marinho.marcia@gmail.com (M.M.M.); helciodossantos@gmail.com (H.S.d.S.); jane.menezes@uece.br (J.E.S.A.d.M.); kirley.canuto@embrapa.br (K.M.C.); 3Departamento de Análises Clínicas e Toxicológicas, Universidade Federal do Ceará, Fortaleza 60714-903, CE, Brazil; tiagosampaio91@gmail.com (T.L.S.); lyanna.rod.rib@gmail.com (L.R.R.); 4Laboratório Farmacologia da Inflamação e do Câncer (LAFICA), Departamento de Farmacologia e Fisiologia, Núcleo de Pesquisa e Desenvolvimento de Medicamentos (NPDM), Universidade Federal do Ceará, Faculdade de Medicina, Fortaleza 60430-275, CE, Brazil; isabelledefatima@gmail.com (I.M.); deysiviviana@ufc.br (D.V.T.W.); 5Laboratório de Fisiologia Endócrina e Metabolismo, Universidade Estadual do Ceará, Campus Itaperi, Fortaleza 60714-903, CE, Brazil; ariclecio.oliveira@uece.br; 6Laboratório de Avaliação e Síntese de Substâncias Bioativas (LASSBio), Instituto de Ciências Biomédicas, Universidade Federal do Rio de Janeiro (UFRJ), Rio de Janeiro 21941-590, RJ, Brazil; lereinag@unal.edu.co (E.R.); lmlima23@gmail.com (L.M.L.); 7Consejo Nacional de Investigaciones Científicas y Técnicas (CONICET), Instituto de Investigaciones en Fisicoquímica de Córdoba (INFIQC), Córdoba X5000HUA, Argentina; walter.pelaez@unc.edu.ar; 8Laboratório de Hormônios e Transdução de Sinais, Departamento de Bioquímica, Centro de Ciências Biológicas, Campus Trindade, Universidade Federal de Santa Catarina, Florianópolis 88040-900, SC, Brazil; mena.barreto@ufsc.br; 9Instituto de Bioeletricidade Celular (IBIOCEL), Ciência & Saúde, Departamento de Bioquímica, Centro de Ciência Biológicas, Universidade Federal de Santa Catarina, Florianópolis 88037-000, SC, Brazil; 10Embrapa Agroindústria Tropical, Fortaleza 60020-181, CE, Brazil

**Keywords:** dipeptyl peptidase-4 inhibitors, insulin resistance, aldose reductase and glucokinase

## Abstract

**Background/Objectives:** Dipeptidyl peptidase-4 (DPP-4) inhibitors are antidiabetic agents that regulate blood glucose by preventing the degradation of active incretin hormones. Although clinically effective, this drug class is associated with adverse effects, creating the need for new molecular scaffolds with improved safety and efficacy. **Methods**: We evaluated the antihyperglycemic activity of β-aminohydrazine and β-amino-N-acylhydrazone derivatives (LASSBio-2123, 2125, 2129, and 2130) using a combined in vivo and in silico approach. Male C57BL/6 mice underwent glucose tolerance tests (GTT) and dexamethasone-induced insulin resistance protocols. Hepatic and skeletal muscle glycogen levels, as well as GLUT4 mRNA expression, were quantified. In silico studies included ADMET predictions and molecular docking analyses against aldose reductase and glucokinase enzymes. MTT was performed on the pancreatic cell line MIN6 (*Mus musculus*). **Results**: Among the compounds tested, LASSBio-2129 demonstrated the most promising profile, with favorable ADMET parameters, metabolic stability, and high docking affinity for aldose reductase and glucokinase. In vivo, LASSBio-2129 (10 mg/kg, i.p.) reduced blood glucose, increased hepatic and muscle glycogen storage, and upregulated GLUT4 mRNA expression in skeletal muscle. Additionally, LASSBio-2129 improved insulin sensitivity in the dexamethasone-induced insulin resistance model, with effects comparable to sitagliptin. **Conclusions**: The combined pharmacological, docking, and ADMET analyses identified LASSBio-2129 as aldose reductase inhibitor candidate and glucokinase activator. Its ability to improve glucose tolerance, enhance glycogen storage, and increase GLUT4 expression highlights its potential as a promising molecule for the treatment of type 2 diabetes mellitus.

## 1. Introduction

Type 2 diabetes mellitus (T2DM) is a chronic, progressive, and multifactorial disease that currently affects more than 500 million adults worldwide, a number projected to reach 783 million by 2045 according to the International Diabetes Federation. The pathophysiology of T2DM involves impaired insulin secretion and sensitivity, hyperglycemia, and metabolic inflexibility, all of which contribute to the development of severe cardiovascular and renal complications [1].

Dipeptidyl peptidase-4 (DPP-4) inhibitors represent a widely used class of oral antidiabetic drugs. By preventing the degradation of the incretin hormones glucagon-like peptide-1 (GLP-1) and glucose-dependent insulinotropic polypeptide (GIP), these agents prolong incretin activity and improve postprandial glucose control [2]. Despite their clinical efficacy, current DPP-4 inhibitors are associated with limitations, including upper respiratory tract adverse events, angioedema risk, and variable patient responses [3,4]. This underscores the need for new molecular scaffolds with optimized safety, efficacy, and pharmacokinetic profiles. Medicinal chemistry efforts have therefore focused on developing novel selective DPP-4 inhibitors, such as β-aminohydrazine and β-amino-N-acylhydrazone derivatives [5], xanthine analogs [6], quinazolinone and pyrimidone analogs [7], β-homophenylalanine analogs [8], and phenethylamine analogs [9] in experimental studies.

On the other hand, verifying whether these new structures bind to additional targets in glucose metabolism is important, as this may represent improvements in more than one aspect of impaired metabolism in diabetic patients [10]. The allosteric activators of the enzyme glucokinase (GK), RO-28-1675 and dorzagliatin, which binds to the allosteric sites of GK, may enhance carbohydrate metabolism in the liver (glycolysis/glycogenesis) and potentiate the insulinotropic response in the pancreas [11]. Still, aldose reductase inhibitors may have beneficial effects in preventing diabetic retinopathy and neuropathy [12]. In particular, computational approaches such as molecular docking, multiparameter optimization (MPO), and absorption, distribution, metabolism, excretion, and toxicity (ADMET) profiling have become essential tools for predicting the pharmacokinetic behavior and toxicity of drug candidates in the early stages of discovery. These in silico strategies, when combined with in vivo assays, accelerate translational drug development and facilitate the identification of lead compounds with high therapeutic potential.

The aim of this study was to evaluate the effect of treatment with 5 iDPP-4 beta-amino hydrazines and beta-amino-acylhydrazones, LASSBio-2123, 2124, 2125, 2129 and 2130, [5] in silico, based on multiparameter optimization (MPO) for absorption, distribution, metabolism, excretion, and toxicity (ADMET) and molecular docking studies in both aldose reductase and glucokinase enzymes. In addition, these five compounds were evaluated in vivo in a glucose tolerance test. From these data, a lead molecule was selected for the other evaluations. LASSBio-2129 was selected to evaluate the hepatic and skeletal muscle glycogen content and insulin sensitization in a dexamethasone-induced insulin resistance model. Treatment with LASSBio-2129 improved glucose tolerance, enhanced glycogen storage in skeletal muscle and liver, and increased GLUT4 mRNA expression, in addition to ameliorating insulin resistance in a dexamethasone-induced model. Furthermore, molecular docking confirmed binding interactions with both aldose reductase and glucokinase enzymes. These findings reinforce the translational potential of LASSBio-2129 and highlight the value of integrating in silico and in vivo approaches in the discovery of next-generation DPP-4 inhibitors for T2DM.

## 2. Results

### 2.1. Effect of β-Aminohydrazines and β-Amino-Acylhydrazones on MPO and ADMET Studies

In a molecular lipophilicity potential (MLP) analysis (Figure 1), it was observed that the aromatic rings of the compounds contributed to the formation of lipophilic surfaces (gray to blue spectrum), whereas aliphatic substructures containing a primary amine and a hydrazide fragment contributed strongly to the formation of polar surfaces (red spectrum), with a topological polar surface area (TPSA) of approximately 67.48 Å^2^ (Table 1). Compound LASSBio-2123, bearing dimethoxy (di-OCH_3_) substitutions, exhibited additive effects on the hydrophilic surface (TPSA = 85.94 Å^2^), resulting in lower lipophilicity (logP = 2.53) (Figure 1A). Conversely, LASSBio-2125 with dichloro (di-Cl) substitutions was affected by the electron-withdrawing inductive effect and by increase in van der Waals surface, leading to high lipophilicity (logP = 3.82) (Figure 1C). This compound showed the lowest MPO score (4.4), reflecting an unfavorable balance between lipophilicity and polarity. Fluorophenyl-based substitutions in LASSBio-2124 (Figure 1B), LASSBio-2129 (Figure 1D), and LASSBio-2130 (Figure 1E) led to optimized van der Waals surfaces and reduced lipophilicity (calculated logP < 3.0), positioning them in a physicochemical space defined by DrugBank^®^ compounds with hydration free energy values ∆G_Hyd_ ≤ −7.0 kcal/mol (Figure 1F). This profile suggests a favorable alignment between lipophilicity and solubility, enhancing ADME attributes.

In the MPO radar analysis (Figure 2A), the control ligand sitagliptin best fit the Pfizer MPO medicinal chemistry criteria (blue region), showing only a single violation (MW > 360 g/mol) and resulting in an MPO score of 5.5. In contrast, LASSBio-2125 showed multiple violations (logP > 3, logD > 2, MW > 360 g/mol), which explained its low MPO score. Among the novel compounds, LASSBio-2129 stood out with an MPO score of 5.3, which was close to that of control drug. This favorable score was attributed to its lower performance in this parameter (AZ logD = 2.32) and reduced basicity (pKa = 6.46), favoring the protonated species R–NH_3_^+^ at physiological pH. In the predicted PAMPA descriptors, fluorinated de- rivatives moderately increased lipophilicity, promoted alignment with high effective cell permeability (P_app,A→B_ MDCK > 1.0 × 10^−6^ cm/s), and had low intrinsic hepatic clearance (Cl_int,u_ < 8.0 mL/min/kg), thereby increasing cell viability. Overall, the compounds showed logD values at a physiological pH between 1.0 and 4.0 and TPSA > 40 Å^2^, which is compatible with oral drug candidates but not typically CNS-active scaffolds.

Alignment of physicochemical descriptors revealed that most LASSBio derivatives reside in an optimized space of P_app,A→B_ and Cl_int,u_, with the exception of LASSBio-2125. This compound exhibited a logD value > 2.8 at pH 7.4 combined with a molecular weight (MW) > 450 g/mol, parameters known to reduce oral bioavailability due to high lipophilicity (Figure 2B). The other analogs (LASSBio-2123, LASSBio-2124, LASSBio-2129, and LASSBio-2130) showed more favorable PAMPA profiles, which we largely attributed to lower lipophilicity (logD < 2.5) (Table 1).

PAMPA predictions based on a machine learning model indicated that logP_app,A→B_ values in Caco-2 cells were concentrated below −5.0, corresponding to permeability rates on the order of 10^−6^ cm/s (Table 2). For MDCK cells, logP_app,A→B_ values clustered between −4.0 and −5.0 and were associated with permeability rates around 10^−5^ cm/s. These results suggest that the compounds permeate more effectively across selective biological membranes, favoring gastrointestinal absorption and distribution across cellular barriers [13]. The lead compound LASSBio-2129 displayed a predicted P_app,A→B_ MDCK of 1.77 × 10^−5^ cm/s, which is slightly lower than sitagliptin (1.99 × 10^−5^ cm/s). However, it showed the largest predicted volume of distribution (Vdss = 7.04 L/kg), suggesting extensive partitioning into adipose tissues [14]. Additionally, Cl_int,u_ values below 7.0 mL/min/kg were predicted for all analogs, indicating that the lipophilicity profile supports slow hepatic clearance without compromising oral bioavailability.

PAMPA descriptors were consistent with high oral bioavailability scores (%F ≥ 0.9, ≈90%), supporting efficient intestinal absorption [15]. Metabolism site analysis indicated that LASSBio-2123 (dimethoxy analogs) could undergo O-dealkylation by CYP3A4, CYP2D6, and CYP2C9, as well as aliphatic hydroxylation of the R_2_C–NH_2_ substructure (41.1 kJ/mol). In halogen-substituted derivatives (LASSBio-2124, LASSBio-2125, LASSBio-2129, and LASSBio-2130), aromatic hydroxylation was predicted, with susceptibility energies of 80.8 kJ/mol, while aliphatic hydroxylation of the R_2_C–NH_2_ group remained predominant (Figure 3).

Compared to sitagliptin, the analogs were considered metabolically safer. Sitagliptin undergoes N-dealkylation of tertiary amines, producing unstable aldehydes that may bind DNA and proteins, contributing to drug-induced liver injury (DILI). AI-based predictions confirmed that fluorinated analogs exhibited enhanced metabolic stability in human and rat hepatocytes, with Cl_Hepa_ < 20 µL/min/10^6^ cells. Microsomal clearance values were also low (<8.0 µL/min/mg), supporting the stability profile predicted by the Pfizer multiparameter optimization system [16,17]. When the physicochemical descriptors were aligned, it was possible to observe that the compounds reside in an optimized space of P_app,A→B_ and Cl_int,u_ with the exception of the derivative LASSBio-2125, where the logD > 2.8at pH 7.4; when aligned with MW > 450 g/mol, it can reduce the oral bioavailability of the compounds due to its high lipophilicity (Figure 2B) resulting from the chloro-substituted aromatic ring. The other compounds, i.e., the analogs LASSBio-2123, LASSBio-2124, LASSBio-2129, and LASSBio-2130, present a more optimized PAMPA profile, especially due to the low lipophilicity (logD < 2.5) (Table 1). Corroborating this, a PAMPA (cell effective permeability) prediction, driven by a machine learning model, showed that the predicted values of logP_app,A→B_ cluster at rates below −5.0 for the Caco-2 cell line (Figure 2C), with permeability rates in the order of 10^−6^ cm/s (Table 2), according to the biopharmaceutical classification system (BCS), with the logP_app,A→B_ MDCK values clustering between −4.0 and −5.0 where the P_app,A→B_ rates are in the order of 10^−5^ cm/s, indicating that the compounds are more permeable in more selective biological membranes, which ensures high absorption in the gastrointestinal tract and better distribution across cell membranes [18,19,20].

Prediction of organic toxicity descriptors indicated that the LASSBio derivatives have a high probability of inducing liver damage (DILI) or Ames mutagenicity due to the potential formation of reactive metabolites. These descriptors were also aligned with a significant probability of rat oral acute (ROA) toxicity and with exceeding the FDA maximum daily dose threshold (FDAMDD ≤ 0.011 mmol/kg-bw/day) [21]. Additionally, the compounds showed a high probability of hepatotoxic responses (H-HT) through metabolic activation, as well as neurotoxicity risk related to their high permeability in the MDCK barrier model, which mimics blood–brain barrier (BBB) penetration (Figure 4A). A Pearson’s correlation-based similarity matrix (Figure 4B) revealed a consistent pattern linking DILI, Ames mutagenicity, hepatotoxicity, and neurotoxicity descriptors. These data collectively suggest that, particularly in cases of overdose, the compounds may have a pronounced tendency to induce toxicity in systemic organs.

### 2.2. Molecular Docking Study on β-Aminohydrazines and β-Amino-Acylhydrazones in Aldose Reductase and Glucokinase Enzymes

After 50 independent docking simulations with 20 poses each, the co-crystallized inhibitors showed binding affinity energy (E_A_) values lower than −6.0 kcal/mol (Figure 5), validating the docking protocol with RMSD < 2.0 Å (Table 3). The LIT inhibitor (aldose reductase) presented an E_A_ of −8.598 kcal/mol, while the 1JD inhibitor (glucokinase) exhibited −7.625 kcal/mol, confirming the reliability of the protocol. All LASSBio analogs displayed E_A_ values below −8.0, indicating high binding affinity for both aldose reductase and glucokinase (Figure 5). Among them, LASSBio-2129 was the most promising, with E_A_ values of −11.085 kcal/mol (aldose reductase) and −10.352 kcal/mol (glucokinase). These values surpassed those of the control ligand sitagliptin, indicating superior interaction energy and a favorable pharmacokinetic profile when aligned with MPO predictions.

By analyzing the complexation of LASSBio analogs in the aldose reductase enzyme, it was possible to observe that the compounds LASSBio-2123, LASSBio-2124, LASSBio-2125, LASSBio-2129, and LASSBio-2130 bound to the cavity of the co-crystallized inhibitor LIT, located in the catalytic site with an estimated molecular surface of 1035.61 Å^2^ (Figure 6A). The inhibitor (LIT) formed essentially hydrophobic interactions with the aromatic portions of the residues Trp79, Phe115, Phe122, and Trp111 (Figure 6B), which are the main residues in the formation of the hydrophobic binding cavity of aldose reductase inhibitors. The LASSBio-2129 ligand complexed to the catalytic site of the enzyme by forming these interactions in common with the inhibitor LIT, including hydrophobic interactions with the aromatic and aliphatic portions of residues Trp79, Phe115, Phe122, and Trp111 (Figure 6B), in addition to weak polar interaction interactions with the polar portion of the residues Asp43 (2.73 Å), Ile260 (3.99 Å), and Cys298 (2.97 Å), with contributions from fluoro-substituted groups, where the donor–acceptor distances (in Å) characterize moderate/weak strength interactions (Table 3) [22]. The control ligand sitagliptin also interacted with this portion of the aldose reductase (Figure 6C), indicating that the compound also acts through this catalytic pathway and constitutes molecular recognition pharmacophores in the discovery of new enzyme inhibitors [23].

In the glucokinase enzyme, the analogs of LASSBio-2123, LASSBio-2124, LASSBio-2125, LASSBio-2129 and LASSBio-2130 complexed with the enzyme in the cavity where the inhibitor 1JD binds, located in the catalytic site, with an estimated molecular surface of 1016.41 Å^2^ (Figure 7A), consisting mainly of apolar aliphatic side chain residues, which include Ile159, Tyr214, Tyr214, Tyr214, Val452, and Val455 (Figure 7B), in addition to aromatic residues responsible for the formation of a hydrophobic portion of the catalytic site, which include Trp99, Tyr214, and Tyr215 (Table 3) [24]. After the simulations, it was possible to observe that the LASSBio-2129 ligand formed hydrophobic interactions with the aromatic portion of the residues Trp99, Tyr214 and Tyr215, and with the apolar portion of Val455 (Figure 7C), while the control ligand interacted more strongly with the residues Trp99 and Tyr214, in addition to forming an H-bond interaction with the nitrogenous portion of Gly68 (Figure 7D).

In an analysis of structural contributions, it was possible to observe that LASSBio-2129 interacted similarly with both aldose reductase and glucokinase (Figure 8). The fluoro-substituted aromatic substructures of the compound contributed strongly to the formation of hydrophobic interactions with the aromatic portions of the Tyr209 and Trp111 residues (Figure 8A), and to the formation of interactions of the same nature with the aromatic residues of Tyr214 and Tyr215 of the glucokinase enzyme (Figure 8B). On the other hand, the primary amine of the compound (R-NH_3_^+^ at pH 7.4) favored the formation of a π-cation nature with the aromatic portion of the Trp20 residue in the aldose reductase enzyme (Figure 8A), while it favored the formation of a polar H-bond donation to the carbonyl portion of Arg63 of the glucokinase enzyme (Figure 8B).

### 2.3. Effect of β-Aminohydrazines and β-Amino-Acylhydrazones on GTT and Glycogen Content

A glucose tolerance test was performed with DPP-4 inhibitors LASSBio-2123, 2124, 2125, 2129, 2130 and sitagliptin (10 mg/kg i.p) in fasting C57Bl/6 mice in Table 4. According to the results, hyperglycemic animals (saline group) showed notable hyperglycemia at times of 15, 30 and 60 min, in relation to time zero, as expected after glucose overload. The sitagliptin group (positive control) reduced blood glucose by 53, 41 and 21% at 15, 30 and 60 min, respectively. LASSBio-2123 and 2129 reduced blood glucose by 18 and 42%, and 56 and 29% at 15 and 30 min, respectively, when compared to the hyperglycemic control group. Furthermore, LASSBio-2130 reduced blood glucose by 24% at 15 min when compared to the hyperglycemic control group. The compounds LASSBio-2124 and 2125 did not reduce blood glucose at any time evaluated at the dose of 10 mg/kg.

For the area under the curve (AUC) for the glycemic curve (ΔAUC_0–120min_), comparing the effect of the impact of treatment with LASSBio-2129 on GTT (Figure 9B), there was a reduction of around 24 and 38% in the AUC of the LASSBio-2129 and sitagliptin (10 mg/kg) treatments, respectively, when compared to the AUC of the hyperglycemic control group (Figure 9B). In our study, the compounds LASSBio-2123 and LASSBio-2129 stood out in the glucose tolerance curve, which serves as a preclinical proof-of-concept test to evaluate antihyperglycemic compounds. In line with a study by Reina et al., 2024 [5], where the inhibition of the DPP-4 enzyme was evaluated in vitro, LASSBio-2123, LASSBio-2129, and sitagliptin presented IC50s of 34.3, 5.08, and 0.092 µM, respectively. In both tests, LASSBio-2129 showed better performance in the regulation of glycemia and in the inhibition of the DPP-4 enzyme. Thus, LASSBio-2129 was chosen for the remaining analyses.

The next step was to evaluate the dose–response curve of the compound, where it was found that the 10 mg/kg dose was more effective in reducing blood glucose than the lower doses (1.0 and 0.1 mg/kg). In the graph of the area under the curve, the higher the bar level, the greater the glycemic response, indicating hyperglycemia, which is undesirable for individuals with diabetes or insulin resistance. Notably, the decrease in AUC induced by LASSBio-2129 administration indicates its potential for being used in the treatment of T2DM, since it did not show a significant difference when compared with the area under the blood glucose curve of the sitagliptin group.

The formation of hepatic and muscular glycogen is beneficial for glucose metabolism, as it indicates a correct insulin signaling pathway and that excess blood glucose is being directed to glycogenesis. The LASSBio-2129 group increased 9.4-fold and 1.86-fold in muscle and liver glycogen contents, respectively, as compared to the hyperglycemic control group. The sitagliptin group only increased its muscle glycogen content by 10.9-fold compared to the hyperglycemic control group in group (Figure 9C,D). Thus, the effect of LASSBio-2129 showed additional beneficial effects to that of sitagliptin at the same dose (10 mg/kg) on liver glycogen content.

### 2.4. Effect of the LASSBio-2129 Compound in Dexamethasone-Induced Insulin-Resistant Mice

Dexamethasone-induced insulin resistance is a good model for assessing insulin resistance in skeletal muscle and adipose tissue [25,26,27]. The control group (saline) presents normal glucose metabolism function, indicating normal insulin sensitivity, with rapid glucose absorption by the tissues. In the dexamethasone group, there was a reduction of approximately 58% in insulin sensitivity compared to the control group, indicating a lower efficiency in glucose uptake in response to insulin (Figure 10A). Treatment with LASSBio-2129 (10 mg/kg, i.p.) increased insulin sensitivity by approximately 127% compared to the dexamethasone group. When LASSBio-2129 was administered to healthy animals, there was no increase in insulin sensitivity compared to the control group (saline), and adverse or toxic effects were not observed during the five days of administration of this compound. Treatment with LASSBio-2129 (10 mg/kg i.p.) increased GLUT4 transporter expression in muscle by 2.77-fold, when compared to the dexamethasone group. LASSBio-2129 in healthy animals did not increase GLUT4 expression when compared to the control group (saline) (Figure 10B).

### 2.5. Effect of the LASSBio-2129 Compound on Cell Viability in MIN6 Cells

LASSBio-2129 reduced MIN6 cell viability in a concentration-dependent manner, with significant effects observed only at 10 µM (≈35% reduction) and 100 µM (≈81% reduction) compared to control cells (Figure 11).

## 3. Discussion

The results of this study indicate that the β-aminohydrazines and β-amino-N-acylhydrazones series, especially LASSBio-2129, exhibited biological performance consistent with a promising antidiabetic profile. In the glucose tolerance model, LASSBio-2129 (10 mg/kg, i.p.) reduced blood glucose and AUC, increased muscle and liver glycogen content, and increased GLUT4 expression in skeletal muscle; additionally, it attenuated dexamethasone-induced insulin resistance, with a magnitude comparable to that of sitagliptin at the same dose. These in vivo effects were in line with favorable in silico predictions (MPO/ADMET) and high binding affinity to aldose reductase (AR) and glucokinase (GK) enzymes. Activation of the glucokinase enzyme can improve carbohydrate metabolism in the liver (glycolysis/glycogenesis) and enhance the insulinotropic response in the pancreas, which may also suggest long-term beneficial effects in the prevention of diabetic neuropathy and retinopathy, as well as in a reduction in hepatic glucose production.

The results demonstrate that aromatic substitutions significantly modulate physicochemical descriptors relevant to drug-likeness and ADMET behavior. The dimethoxy group in LASSBio-2123 increased polarity and reduced lipophilicity, favoring a better balance between solubility and permeability, in agreement with established medicinal chemistry profiles [18,28]. In contrast, LASSBio-2125 displayed high lipophilicity and a poor MPO score, a combination typically linked to metabolic instability and off-target toxicity [29,30]. Fluoro-phenyl substitution emerged as a rational optimization strategy. LASSBio-2124, LASSBio-2129, and LASSBio-2130 demonstrated improved lipophilicity–permeability balance, consistent with prior studies highlighting fluorination as a key medicinal chemistry tool to enhance bioavailability and metabolic resistance [30,31]. Among these, LASSBio-2129 achieved an MPO score (5.3) similar to sitagliptin, strengthening its candidacy as a lead compound.

The alignment of predicted descriptors with favorable ADMET parameters (high permeability, low clearance, and oral bioavailability ≈ 90%) reinforces that these derivatives have the potential for efficient intestinal absorption and metabolic stability. These findings underscore the value of rational substituent design—particularly fluorination—in generating drug-like scaffolds with optimized profiles suitable for preclinical prioritization [18]. Within this context, LASSBio-2129 stood out by combining adequate permeability with a higher Vdss, suggesting broader tissue distribution, which could improve systemic exposure. Its pharmacokinetic attributes, coupled with balanced clearance, indicate translational potential [14].

Metabolism predictions further support the relevance of structural modifications in defining susceptibility to hepatic transformation. While methoxy groups favored classical O-dealkylation, halogenated analogs underwent aromatic hydroxylation, a more stable metabolic pathway. Importantly, unlike sitagliptin, the LASSBio derivatives did not generate unstable aldehydes, which are associated with hepatotoxicity and idiosyncratic drug-induced liver injury [32,33]. Fluoro-substituted derivatives were particularly promising for their metabolic stability, consistently showing low clearance in both hepatocyte and microsomal assays. These results reinforce the protective role of fluorine in modulating metabolic resistance and preventing rapid elimination. Collectively, these findings support prioritization of fluorinated analogs, especially LASSBio-2129, for progression into advanced preclinical development. Nevertheless, predictions of hepatotoxicity, mutagenicity, and neurotoxicity require careful interpretation and experimental validation. To address these safety concerns, we have now included experimental data on cell viability using the MTT assay in pancreatic MIN6 β-cells (Figure 11). LASSBio-2129 did not significantly reduce cell viability at concentrations up to 1 µM, whereas reductions of approximately 35% and 81% were observed only at higher concentrations (10 µM and 100 µM, respectively). These findings indicate that the compound is well tolerated at pharmacologically relevant low micromolar concentrations, while cytotoxicity emerges only at supra-pharmacological doses. This experimental evidence provides an initial indication of safety, complementing the ADMET predictions, although more extensive in vitro and in vivo toxicological studies remain necessary.

Nevertheless, safety predictions demand careful consideration. The elevated probability of DILI and Ames mutagenicity observed for the class is consistent with the literature describing the metabolic activation of hydrazone-like scaffolds into electrophilic intermediates capable of damaging proteins and DNA [34,35]. The neurotoxicity risk, potentially linked to blood–brain barrier penetration, underscores the importance of minimizing CNS exposure in molecules not intended for central action [18]. Mechanistic pathways such as oxidative stress, mitochondrial dysfunction, and covalent adduct formation may underlie these liabilities, emphasizing the need for further optimization [36]. Docking simulations confirmed that the designed compounds, particularly LASSBio-2129, interact strongly with aldose reductase (AR) and glucokinase (GK). These two enzymes are highly relevant therapeutic targets: AR participates in the polyol pathway and contributes to microvascular complications of diabetes, whereas GK functions as a glucose sensor in liver and β-cells, influencing glucose homeostasis [37,38]. LASSBio-2129 demonstrated superior binding compared to sitagliptin, largely due to favorable weak polar interactions, with a strong contribution from the fluorine-substituted groups, and π-cation interactions, which are known to stabilize drug–target complexes [39,40]. Its interactions with conserved pharmacophoric residues of AR (Trp79, Phe115, Trp111,and Tyr48) and GK (Trp99, Tyr214, and Tyr215) recapitulate motifs observed in known potent inhibitors [37,38].

The E_A_ values lower than –8.0 observed for the analogs of LASSBio-2123, LASSBio-2124, LASSBio-2125, LASSBio-2129, and LASSBio-2130 reveal the high affinity of the compounds for the catalytic pathways of aldose reductase and glucokinase (Figure 5), resulting from a high structural specificity in relation to sitagliptin [41]. Here, the compound LASSBio-2129 stands out, which presented an alignment between a viable pharmacokinetic profile by MPO and lower E_A_ rates in relation to the other compounds, with calculated values in the order of −11,085 and −10,352 kcal/mol, against aldose reductase and glucokinase, respectively (Figure 5), as indicators of ideal pharmacokinetic and pharmacodynamic properties [42]. Dual modulation of AR and GK offers a therapeutic advantage, combining potential protection against diabetic complications (via AR inhibition) with improved glucose utilization (via GK activation). This multitarget activity positions LASSBio-2129 as a strong lead with complementary mechanisms of action.

Integration of MPO and ADMET analyses further supports the compound’s classification within a favorable physicochemical window—logD ≈ 2 and TPSA 60–90 Å^2^—consistent with oral bioavailability. This aligns with evidence that moderate lipophilicity and balanced polarity enhance absorption and reduce metabolic liabilities when properly optimized [18,28]. The docking analysis of AR revealed that the active site consists of anion-binding (Tyr48, His110, and Trp111) and hydrophobic specificity pockets (Trp20, Val47, Phe115, and Phe122), and LASSBio-2129 fitted within these cavities with high theoretical affinity, in line with prior reports of AR inhibitors [37]. Although AR inhibition may not fully explain acute glucose tolerance improvements, chronic modulation of AR combined with DPP-4 inhibition could mitigate osmotic and oxidative stress, complementing glycemic control. Finally, the in vivo results provide strong translational evidence. LASSBio-2129 reduced glycemia at the same pharmacological dose as sitagliptin, corroborating prior findings with synthetic analogs [43]. Its effects on hepatic and muscular glycogen storage and GLUT4 expression align with DPP-4 inhibition pathways that enhance incretin half-life and promote glucose uptake [7,9]. The improvement in insulin resistance in the dexamethasone model, a validated paradigm for testing insulin sensitizers [44], reinforces the pharmacological relevance of LASSBio-2129. Taken together, these findings demonstrate that LASSBio-2129 combines in silico. The original contributions presented in this study are included in the article. Further inquiries can be directed to the corresponding author(s), and in vivo properties consistent with a promising antidiabetic candidate. Its multitarget profile, favorable pharmacokinetics, and observed improvements in insulin sensitivity and GLUT4 expression justify its prioritization for further preclinical development.

The MPO and ADMET data place LASSBio-2129 in a physicochemical space characterized by good permeability (Caco-2/MDCK), which is a classical attribute of candidates with the best chance of adequate oral exposure. The usefulness of multiparameter scores (Pfizer MPO) to align lipophilicity (logP/logD), TPSA, pKa, and MW with favorable ADME profiles has been widely demonstrated, and recent reviews have reinforced that optimizing lipophilicity within narrow windows improves permeability without compromising metabolism (as long as metabolic soft spots are simultaneously addressed). Our fluorinated compounds, with logD ~2 and TPSA in the 60–90 Å^2^ range, fit within this equilibrium window [28]. Although permeability predictions (PAMPA/Caco-2/MDCK) and Vdss/Cl_int,u_ are not a substitute for in vivo pharmacokinetics, the observed direction (P_app_ in the range of 10^−6^–10^−5^ cm/s and low Cl_int,u_) is consistent with high intestinal absorption and metabolic stability, described as desirable in oral drugs [18].

Aldose reductase (AR) is part of the polyol pathway and is implicated in diabetes complications; its active site contains an anion-binding pocket (Tyr48, His110, and Trp111) and a hydrophobic region (specificity pocket) that includes Trp20, Val47, Phe115, and Phe122 [37,45]. Our simulations positioned LASSBio-2129 in cavities that recapitulated hydrophobic and weak polar interactions described for model inhibitors, supporting high theoretical affinity. Although AR inhibition does not directly explain the acute GTT improvement, the polytarget (iDPP-4 ± AR) may add value in chronic settings by mitigating osmotic and oxidative stress.

LASSBio-2129 reduced blood glucose at the same pharmacological dose as the sitagliptin group. Earlier, our research group demonstrated the blood glucose-lowering effect of new synthetic compounds such as the chalcone analogs (E)-3-(phenyl)-1-(3,4,5-trimethoxyphenyl)prop-2-en-1-one (10 mg/kg, i.p.) [46], using fasted Wistar rats in the GTT. Furthermore, the effect of the LASSBio-2129 on glucose homeostasis corroborates previous research involving selective iDPP-4 acting to increase the half-life of endogenous GLP-1 and GIP [6,7,8,9]. In addition, administration of LASSBio-2129 (10 mg/kg) increased hepatic and muscular glycogen content, which contributes to the reduction in glycemia and the insulinomimetic effects of this compound. Other chemically synthesized compounds also increase glycogen formation [27,46,47].

There is evidence that hepatic DPP-4 activity/expression is increased in conditions of glucocorticoid-induced insulin resistance [48,49]. Furthermore, the dexamethasone insulin resistance model has been widely used to test insulin-sensitizing agents, with 5–7-day protocols reproducing impaired peripheral glucose uptake and worsening tolerance [27,44]. The insulin resistance-attenuating effect of LASSBio-2129 can be generated through daily glycemic control by inhibiting the DPP-4 enzyme, which is known to prolong the half-life of GLP-1 and GIP, thereby corroborating the effect observed in this study.

The improvement in Kitt and the glycemic curve after LASSBio-2129 is in line with what is described for reference agents in this model, reinforcing the validity of the finding. In skeletal muscle, GLUT4 is crucial for insulin-stimulated glucose uptake; increases in its expression and/or translocation are associated with improved insulin sensitivity and glucose tolerance. DPP-4 enzyme activity has different modes of action in glucose metabolism. It has been found that upregulation of hepatic DPP-4 ex-pression is likely the cause of glucose intolerance or insulin resistance [50]. In a clinical study, allogliptin, an iDPP-4, modulated insulin resistance and atherogenic lipid metabolism in adipose tissue [51]. Saxagliptin (8mg/kg), an iDPP-4, ameliorated the dexamethasone-induced hyperglycemia and insulin resistance in female Wistar rats [52]. Combining the data obtained, it was found that the compound LASS-Bio-2129 reduced glycaemia and insulin resistance, in addition to increasing the muscle glycogen and GLUT4 expression. The increase in GLUT4 mRNA observed with LASSBio-2129 is therefore consistent with the improvement in GTT and insulin sensitivities. There are even reports that DPP-4 inhibition can increase GLUT4 in muscle/heart via GLP-1-dependent signaling, suggesting an incretin-muscle axis that may have contributed to the effects observed here [53]. These findings demonstrate that the drug under investigation promoted an increase in the sensitivity of muscle tissue to the action of insulin, probably due to the increase in glucose uptake by skeletal muscle. Furthermore, LASS-Bio-2129, by itself, increased the expression of GLUT4, which favors glucose metabolism and contributes to glycemic control.

## 4. Materials and Methods

### 4.1. Chemical Synthesis of the Dipeptyl Peptidase-4 Inhibitors

The (R,E)-3-amino-N’-(3,4-difluorobenzilideno)-4-(2,4,5-trifluorofenil)butanohidrazida (LASSBio-2129) and others ^®^-amino-N-acylhydrazones (LASSBio-2123, LASSBio-2124, LASSBio-2125, and LASSBio-21230) (Figure 1) evaluated in this study were originally synthesized and described by Reina et al., 2024 [5].

### 4.2. Multiparameter Optimization-Based Absorption, Distribution, Metabolism, Excretion, and Toxicity (ADMET) Studies

#### 4.2.1. Lipophilicity Potential and Hydration Free Energy

The two-dimensional representation of the chemical structure of the ligands was plotted and rendered in the academic license software MarvinSketch^®^ version 25.1.0, Chemaxon© (https://chemaxon.com/marvin, accessed: 24 August 2025) for plotting the surface map of the Molecular Lipophilicity Potential (MLP), as shown in Equation (1):(1)$${MLP}_{k}=\sum_{i=1}^{N}\;f_{i}.F\left(d_{ik}\right)$$
where *N* is the number of fragments (*i*) with lipophilicity constant *f_i_*, and *F* is the distance function (*d_ik_*) between each fragment [54]. The results were related to the descriptors of intrinsic lipophilicity (*logP*), lipophilicity at physiological pH (logD at pH 7.4), topological polar surface area (TPSA), and free energy of hydration (*ΔG_hyd_*), which was calculated from its relationship with logP, using the artificial intelligence (AI)-based absorption, distribution, metabolism, excretion, and toxicity (ADMET) property predictor ADMET-AI (https://admet.ai.greenstonebio.com/, accessed: 24 August 2025), as shown in Equation (2) [55]:(2)$$logP=\frac{1}{2.303RT}\left({\Delta G}_{hyd}\right)$$

Then, the calculated physicochemical properties of the derivatives were converted into quantitative druglikeness estimation scores using the Multiparameter Optimization (MPO) algorithm, as shown in Equation (3):(3)$$D=\sum_{k=1}^{N}\;w_{k}T_{k}\left(x_{k}^{0}\right)$$
where *w* is the weighting factor of each physicochemical attribute *k* in relation to the distance functions of the thresholds (*T*(*x*)) formed by logP ≤ 3, logD at pH 7.4 ≤ 2, molecular weight (MW) ≤ 360 g/mol, TPSA 40-90 Å^2^, H-bond donors (HBD) ≤ 1, and pKa (basic) ≤ 8 (N = 6), where the sum results in a score that varies from 0.0 to 6.0 according to the pharmacokinetic viability [56].

#### 4.2.2. PAMPA Predicted Descriptors

The AI-guided similarity test was configured to estimate the Parallel Artificial Membrane Permeability Assay (PAMPA) properties for P_app,A→B_ in human colon epithelial cells (Caco-2) and Madin–Darby canine kidney (MDCK) cell lines through a regression model that uses quantitative structure–property relationship (QSPR) data, resulting in logPapp values being filtered by the biopharmaceutical classification system, where compounds with P_app,A→B_ > 20 × 10^−6^ cm/s present high cell permeability and a human intestinal absorption of around 96% [13]. The values were related to the prediction of oral bioavailability and volume of distribution of the steady state (Vdss) using the ADMET-AI (https://admet.ai.greenstonebio.com/,accessed: 24 August 2025) and ADMETlab 3.0 (https://admetlab3.scbdd.com/,accessed: 24 August 2025) predictors [42].

#### 4.2.3. Metabolic Stability and Toxicity Prediction

The prediction of metabolic stability was made through the similarity test, based on linear regressions, with compounds present in the ChEMBL database (https://www.ebi.ac.uk/chembl/explore/document/CHEMBL3301361, accessed: 24 August 2025) with data from in vitro pharmacokinetic and metabolic stability assays. The properties include lipophilicity at physiological pH (logD at pH 7.4) and clearance descriptors in liver microsome (Cl_Micro_) and hepatocyte (Cl_Hepa_) systems (human liver microsomes, and human and rat hepatocytes) where compounds with Cl_Hepa_ < 100 µL/min/10^6^ cells (Cl_int,u_ < 100 mL/min/kg) are associated with metabolic stability values. Metabolism site prediction was performed using the Site of Metabolism prediction for Cytochrome P450s (SMARTCyp) tool (https://smartcyp.sund.ku.dk/mol_to_som, accessed: 24 August 2025), generating a relative susceptibility map of compounds as substrates of CYP450 isoforms. The results were compared with the predicted probability of compounds resulting in an organic toxic response using the ADMETlab 3.0 server (https://admetlab3.scbdd.com/, accessed: 24 August 2025), and statistically analyzed using Morpheus^®^ web server (https://software.broadinstitute.org/morpheus/, accessed: 24 August 2025).

### 4.3. Molecular Docking Study

To estimate the theoretical mechanism against T2DM, the ligand–protein interactions between the derivatives LASSBio-2123, LASSBio-2124, LASSBio-2125, LASSBio-2129 and LASSBio-2130 and the enzymes aldose reductase and glucokinase were investigated [24]. The chemical structure of “Human aldose reductase complexed with nitrofuryl-oxadiazole inhibitor at 1.55 A” and “Crystal structure of Human Glucokinase in complex with a small molecule activator” were taken from the RCSB Protein Data Bank repository (https://www.rcsb.org/, accessed: 24 August 2025), deposited under PDB ID codes 2IKH and 4IXC, respectively, both in *Homo sapiens* organisms and the *Escherichia coli* expression system, and resolved by X-ray diffraction at a resolution of 1.55 and 2.00 Å, respectively. For preparation of the enzymes, water molecules (H_2_O) were removed, hydrogens were added, and Gasteiger charges were computed using AutoDockTools^TM^ software, version 1.5.7 (https://autodocksuite.scripps.edu/adt/, accessed: 24 August 2025) [57]. Then, the software was used to configure the grid-box, which was adjusted under the axes x = 14.17, y = −0.095 and z = 23.464 and dimensions 60x40x56 Å, for the enzyme aldose reductase, and under the axes x = 63.267, y = 25.856 and z = 0.231 and dimensions 66x72x66 Å, for the enzyme glucokinase. Then, the AutoDockVina^TM^; code, version 1.2.x (https://vina.scripps.edu/, accessed: 24 August 2025) was executed to perform a series of 50 independent molecular docking simulations, of 20 poses each, using the Lamarckian Genetic Algorithm (LGA) and degree of exhaustiveness = 64, whose best-pose selection criterion included affinity energy (EA) lower than −6.0 kcal/mol, within the ideal statistical threshold formed by a Root Mean Square Deviation (RMSD) lower than 2.0 Å [58].

### 4.4. Animals

Male C57BL/6 mice (7 weeks, 10–20 g) were kept in cages at 21 ± 2 °C and a light-dark cycle of 12 h (light on between 6 and 6 h) with feed (Nuvital, Curitiba, PR, and Brazil) and water ad libitum. The experiments were carried out in the bioterium of the Center for Research and Development of Medicines (NPDM) at Federal University of Ceará according to the Ethics Committee on Animal Use (CEUA-NPDM/UFC: Nº 17010720-0). This bioterium is certified by the Association for Assessment and Accreditation of Laboratory Animals (AAALAC).

### 4.5. Effects of LASSBio-2123, 2124, 2125, 2129 and 2130 on Glucose Tolerance Test

Fasted mice (6 h) were divided into groups of 7 rats for the glucose tolerance test: hyperglycemic (received 2 g glucose/kg of body weight plus vehicle, i.p.); hyperglycemic plus LASSBio 2129 (0.1, 1, and 10 mg/kg, i.p.) or sitagliptin (10 mg/kg, i.p.). Glycemia was measured before any treatment or glucose overload (zero time represents fasting glycemia). LASSBio-2123, 2124, 2125, 2129, 2130, and sitagliptin, and glucose (2 g/kg of body weight) was administered 30 min later. Glycemia was measured at 15, 30, 60, and 180 min after glucose overload by the glucose oxidase method [59].

### 4.6. Glycogen Content Measurements

Glycogen was isolated from the liver and soleus muscle of the animals at 120 min in the GTT assay. The tissues were weighed, homogenized in 33% KOH, and boiled at 100 °C for 20 min, with occasional stirring. After cooling, 96% ethanol was added to the samples and heated to boiling followed by cooling in an ice bath to aid the precipitation of glycogen. The homogenates were centrifuged at 3000 rpm for 15 min, the supernatant was discarded. Glycogen content was determined by treatment with iodine reagent and the absorbance was measured at 460 nm [60]. The results were expressed as mg of glycogen/g of tissue.

### 4.7. Insulin Resistance and Insulin Tolerance Test

To evaluate the effect of cashew nut oil on insulin resistance, mice were divided into four groups: (1) mice received vehicle (saline), (2) 0.1 mg/kg dexamethasone subcutaneously (s.c.), (3) LASSBio-2129 10 mg/kg; i.p. and (4) LASSBio-2129 10 mg/kg; i.p. plus dexamethasone (0.1 mg/ kg s.c.). The mice were induced with daily subcutaneous injections from 8 h 30 min to 9 h 30 min a.m., for 5 consecutive days [59].

Mice that had been fasted for at least 4 h received subcutaneously injected insulin (2 U/kg body weight). Blood was collected from the tail at 0, 7, 14, and 28 min to determine glucose concentrations. This test measures insulin sensitivity using the constant disappearance of glucose calculated using the following formula: K_itt_ = 0.693 × 100/t_½_. where t_½_ is the half-life of decay of glucose and was determined from the slope of the line obtained by linear regression of the natural logarithm of glucose versus time [61].

### 4.8. Real-Time PCR

Muscle samples from mice treated or not treated with dexamethasone were ground and homogenized to RNA isolation, for which the TRIzol^TM^; Reagent (Invitrogen, Carlsbad, CA, USA) using steel beads (4.5 mm) that were shaken in the TissueLyser LT (Qiagen, Hilden, Germany). We assessed the yield and quality of total RNA using the Epoch^TM^; Microplate Spectrophotometer (BioTek Instruments, Winooski, VT, USA). RNA reverse transcription was performed using High-Capacity cDNA Reverse Transcription Kit (Applied Biosystems, Thermo Fisher Scientific, Waltham, MA, USA) and Veriti^TM^; 96-Well Thermal Cycler (Applied Biosystems, Thermo Fisher Scientific, Waltham, MA, USA). The real-time PCR amplification was carried out using 2 μL of cDNA, specific primers for each gene, and SyBR Green reagent (Invitrogen, Carlsbad, CA, USA), in a final volume of 10 μL. The 2−(ΔΔCt) method was used to calculate the ΔΔCt values. Real-time PCR was accomplished on QuantStudio^TM^; 3 equipment (Thermo Fisher Scientific, Waltham, MA, USA) [62]. The primer sequences used were as follows: GLUT4: forward:5′-CGCGGCCTCCTATGAGATAC-3′; reverse: 5′-CCTGAGTAGGCGCCAA TGA-3′. 

### 4.9. Cell Culture and Viability Assays

MIN6 cells derived from the mouse pancreas (*Mus musculus*) were cultured in low-glucose Dulbecco’s Modified Eagle’s Medium (DMEM; Gibco, Thermo Fisher Scientific, Waltham, MA, USA) supplemented with 2 mM L-glutamine, 1% penicillin–streptomycin (GE Healthcare, Chicago, IL, USA), and 10% fetal bovine serum (FBS; Gibco, Thermo Fisher Scientific, Waltham, MA, USA).

Cell viability was assessed using the 3-(4,5-dimethylthiazol-2-yl)-2,5-diphenyltetrazolium bromide (MTT) assay. MIN6 cells were treated with LASSBio-2129 for 24 h, after which MTT was added to the culture medium at a final concentration of 2.5 mg/mL and incubated at 37 °C for 4 h. Inside viable cells, MTT is reduced to insoluble formazan crystals, which were subsequently solubilized by adding 10% sodium dodecyl sulfate (SDS). After 17 h of incubation, absorbance was measured at 570 nm using a microplate reader [63]. Initially, the MTT assay was performed to determine the cytotoxicity range of LASSBio-2129 in MIN6 cells. The assay was then applied to evaluate cell death following the ischemia/reoxygenation protocol and to assess the recovery capacity of MIN6 cells in the presence of LASSBio-2129.

### 4.10. Data and Statistical Analysis

Data are expressed as means ± S.E.M. An analysis of variance (ANOVA) followed by Bonferroni post-test or Student’s *t* test was used to determine the significance of differences.

## 5. Conclusions

MPO-based ADMET estimates classified LASSBio-2129 as a leading compound due to its low lipophilicity at physiological pH and alignment between high cell permeability and metabolic stability in liver microsomes, although it is estimated to be a pharmacological active principle based on the control of the daily oral dose administered. Molecular docking simulations showed that the analogs can bind to the catalytic site of both aldose reductase and glucokinase in the treatment of T2DM, with emphasis on LASSBio-2129 due to its low affinity energy and high pharmacokinetic viability. The effects of LASSBio-2129 (10 mg/kg) on a glucose tolerance test and insulin sensitivity on insulin resistance were partially mediated by increased GLUT 4 expression in muscle tissues. Still, LASSBio-2129 was safe up to 10 µM in the cell viability test in MIN6. The data set obtained shows that LASSBio-2129 may have similar effects to sitagliptin on glucose metabolism and possibly fewer adverse effects, it a promising molecule for the treatment of diabetes.

## Data Availability

The original contributions presented in this study are included in the article. Further inquiries can be directed to the corresponding author(s).

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
