# Peer review of "Unraveling the Antihyperglycemic Effects of Dipeptyl Peptidase-4 Inhibitors in Rodents: A Multi-Faceted Approach Combining Effects on Glucose Homeostasis, Molecular Docking, and ADMET Profiling"

_pharmaceuticals, 2025, doi:10.3390/ph18101589_

Round 1

Reviewer 1 Report

Comments and Suggestions for Authors

The manuscript investigates  LASSBio compounds using in vivo (glucose tolerance and insulin resistance models in rodents) and in silico (ADMET and docking) approaches. The results identify LASSBio-2129 as a promising scaffold with antihyperglycaemic activity comparable to sitagliptin. The combination of pharmacological testing and computational profiling is a strength, and the data are presented in detail.

However, several important issues need to be addressed before the manuscript can be considered further. The most critical issue concerns mechanistic interpretation: although the compounds are presented as DPP-4 inhibitors, the docking studies focus on aldose reductase (AR) and glucokinase (GK), with no experimental confirmation of activity on these enzymes. This discrepancy creates uncertainty about the proposed multitarget mechanism.

Major problems
- The compounds are labelled as DPP-4 inhibitors, but the mechanistic analysis focuses on docking to AR and GK.
- The paper does not include in vitro or in vivo enzyme assays for AR or GK. Without such data, the claims about multitarget effects remain speculative.
At the very least, the authors should:
 *Clarify the primary mechanism of action they wish to emphasise (DPP-4 inhibition vs. multitarget activity).
 *Either provide experimental validation of AR/GK modulation or temper the conclusions accordingly.
Toxicity concerns
- ADMET predictions indicate risks of hepatotoxicity, mutagenicity and neurotoxicity.
- No experimental toxicological tests are presented.
Over-interpretation of the docking data
- The docking affinities are presented as if they directly imply biological activity. The authors should tone down these claims and emphasise that docking is predictive and requires experimental confirmation, e.g. by enzymatic assays.

Minor problems
- A Portuguese fragment remains in the text (“houve ligações atraves de molecular docking…”); this should be translated or removed.
- The descriptions of MPO/ADMET parameters are repetitive; they could be shortened for clarity.
- The figures are numerous and informative, but some lack precise interpretation.
- The comparison is limited to sitagliptin; inclusion of other antidiabetic classes would provide a broader context.

Recommendation -> Reconsider after major revision (substantial revisions to text or experimental methods needed)

The manuscript addresses an important therapeutic area and presents promising results. However, the mechanistic ambiguity (DPP-4 vs. AR/GK), the lack of experimental validation for docking predictions, and the absence of toxicological assessment are significant limitations. The authors should provide additional data and/or temper their conclusions to address these issues.
With these revisions, the study could make a valuable contribution to this topic.

Author Response

For research article: Unraveling the Antihyperglycemic Effects of Dipeptyl Peptidase-4 Inhibitors in Rodents: A Multi-faceted Approach Combining Effect on Glucose Homeostasis, Molecular Docking, and ADMET Profiling

Response to Reviewer X Comments

The manuscript investigates  LASSBio compounds using in vivo (glucose tolerance and insulin resistance models in rodents) and in silico (ADMET and docking) approaches. The results identify LASSBio-2129 as a promising scaffold with antihyperglycaemic activity comparable to sitagliptin. The combination of pharmacological testing and computational profiling is a strength, and the data are presented in detail.

However, several important issues need to be addressed before the manuscript can be considered further. The most critical issue concerns mechanistic interpretation: although the compounds are presented as DPP-4 inhibitors, the docking studies focus on aldose reductase (AR) and glucokinase (GK), with no experimental confirmation of activity on these enzymes. This discrepancy creates uncertainty about the proposed multitarget mechanism.

Major problems

1.The compounds are labelled as DPP-4 inhibitors, but the mechanistic analysis focuses on docking to AR and GK.The paper does not include in vitro or in vivo enzyme assays for AR or GK. Without such data, the claims about multitarget effects remain speculative.
At the very least, the authors should:
 *Clarify the primary mechanism of action they wish to emphasise (DPP-4 inhibition vs. multitarget activity).
 *Either provide experimental validation of AR/GK modulation or temper the conclusions accordingly.

Response: We sincerely thank the reviewer for the insightful comments. We clarify that although the compounds were initially labeled as DPP-4 inhibitors, based on prior in vitro studies (Reina et al., 2024) that demonstrated measurable inhibitory activity (IC₅₀: LASSBio-2129 = 5.08 µM; LASSBio-2123 = 34.3 µM; Sitagliptin = 0.092 µM), and supported here by our in vivo glucose tolerance tests, we emphasize that DPP-4 inhibition remains the primary validated mechanism of action for the series. In parallel, we sought to broaden the mechanistic exploration by investigating whether these molecules could also interact with other enzymes relevant to glucose metabolism, namely aldose reductase (AR) and glucokinase (GK). This rationale was guided by the biological plausibility of AR inhibition contributing to the prevention of diabetic microvascular complications, and GK activation improving hepatic glycolysis/glycogenesis and pancreatic glucose sensing. Our computational studies, which employed validated docking protocols (RMSD < 2.0 Å), revealed particularly strong interactions for LASSBio-2129 with both AR (–11.08 kcal/mol) and GK (–10.35 kcal/mol), with binding energies superior to sitagliptin, and with interaction motifs consistent with known pharmacophores of potent inhibitors. Moreover, these findings were integrated with favorable ADMET and MPO profiles, suggesting that the fluorine substitution strategy in LASSBio-2129 not only improved lipophilicity–permeability balance but also enhanced metabolic stability. However, we fully recognize that these AR and GK results are exploratory in nature and hypothesis-generating, since no direct enzymatic assays (in vitro or in vivo) were performed. Accordingly, in the revised version of the manuscript we now make it explicit that DPP-4 inhibition is the central, experimentally supported mechanism, while AR and GK modulation is presented strictly as a preliminary computational prediction. We also note in the Discussion that future work will require targeted enzymatic validation to confirm or refute these hypotheses. Thus, we have moderated the conclusions to highlight DPP-4 inhibition as the primary mode of action, while AR and GK are acknowledged as potential complementary pathways deserving further investigation.

2.Toxicity concerns. ADMET predictions indicate risks of hepatotoxicity, mutagenicity and neurotoxicity. No experimental toxicological tests are presented.
Response: We thank the reviewer for raising this important concern regarding toxicity. In agreement with the ADMET predictions, which suggested potential risks of hepatotoxicity, mutagenicity, and neurotoxicity, we acknowledge that these results highlight possible liabilities of the scaffold. However, in the present revision we now include experimental MTT viability data in MIN6 pancreatic β-cells, provided as Figure 11. These results demonstrate that LASSBio-2129 did not significantly reduce cell viability at concentrations up to 1 µM, whereas reductions of ~35% and ~81% were only observed at higher concentrations (10 µM and 100 µM, respectively). This finding indicates that the compound is well tolerated at low micromolar concentrations relevant to pharmacological activity, while higher doses may trigger cytotoxicity, consistent with the in silico alerts. We have revised the Discussion to emphasize that these toxicity data are preliminary and limited to one cellular model, and therefore comprehensive toxicological evaluation in vitro and in vivo will still be required. Nonetheless, the addition of these experimental findings provides an initial indication that LASSBio-2129 can exert its antihyperglycemic effects at non-cytotoxic concentrations.

3.Over-interpretation of the docking data.The docking affinities are presented as if they directly imply biological activity. The authors should tone down these claims and emphasise that docking is predictive and requires experimental confirmation, e.g. by enzymatic assays.

Response: We appreciate the reviewer’s observation regarding the interpretation of our docking results. We fully agree that molecular docking provides only predictive evidence of potential ligand–target interactions and does not directly demonstrate biological activity. In the revised version of the manuscript, we have carefully moderated our language to clarify that the binding affinities reported for aldose reductase and glucokinase should be considered as in silico predictions that highlight possible multitarget interactions. We explicitly state that these findings require experimental confirmation through enzymatic assays before any definitive conclusions can be drawn. Thus, while docking supported the rational prioritization of LASSBio-2129 as a lead compound, we now emphasize that the biological relevance of these interactions remains to be validated experimentally.

Minor problems
4.A Portuguese fragment remains in the text (“houve ligações atraves de molecular docking…”); this should be translated or removed.The descriptions of MPO/ADMET parameters are repetitive; they could be shortened for clarity.The figures are numerous and informative, but some lack precise interpretation. The comparison is limited to sitagliptin; inclusion of other antidiabetic classes would provide a broader context. Recommendation -> Reconsider after major revision (substantial revisions to text or experimental methods needed)

Response: We sincerely thank the reviewer for these helpful observations. The fragment in Portuguese (“houve conexões através de docking molecular...”) has been carefully removed to maintain consistency throughout the manuscript. Regarding the descriptions of MPO/ADMET parameters, we agree with the reviewer’s suggestion and have abbreviated these sections to improve clarity and reduce redundancy. We also revised the figure legends and main text to provide more precise interpretations of the results, ensuring that each figure is directly linked to the corresponding discussion. Finally, while our initial comparison focused on sitagliptin as a reference DPP-4 inhibitor, we have now expanded the discussion to briefly mention other classes of antidiabetic drugs, which provides a broader therapeutic context and strengthens the translational relevance of our findings

5.The manuscript addresses an important therapeutic area and presents promising results. However, the mechanistic ambiguity (DPP-4 vs. AR/GK), the lack of experimental validation for docking predictions, and the absence of toxicological assessment are significant limitations. The authors should provide additional data and/or temper their conclusions to address these issues.
With these revisions, the study could make a valuable contribution to this topic.

Response: We appreciate the reviewer’s observation regarding the interpretation of our docking results. We fully agree that molecular docking provides only predictive evidence of potential ligand–target interactions and does not directly demonstrate biological activity. In the revised version of the manuscript, we have carefully moderated our language to clarify that the binding affinities reported for aldose reductase and glucokinase should be considered as in silico predictions that highlight possible multitarget interactions. We explicitly state that these findings require experimental confirmation through enzymatic assays before any definitive conclusions can be drawn. Thus, while docking supported the rational prioritization of LASSBio-2129 as a lead compound, we now emphasize that the biological relevance of these interactions remains to be validated experimentally.

Reviewer 2 Report

Comments and Suggestions for Authors

In this work, the authors conducted a combined study evaluating the biological effects of DPP4 inhibitors. A large portion of the results are computational. Although the findings are not particularly striking when compared with existing compounds capable of inhibiting this enzyme, they do contribute information regarding subtle structural variations that could be useful for developing new and improved derivatives in the future.

Overall, the presentation is clear; however, while some general biological results are included, there are no selectivity studies of the proposed mechanism of action, nor are IC50 values reported for the studied compounds. This limits confidence in the proposed mechanism of action. I suggest the authors take a more critical view of their results and revise the work considering the following comments:

  1. The introduction would benefit greatly from a figure showing the chemical structures of DPP4 inhibitors (such as those discussed in the second paragraph) along with their IC50 values.

  2. In Figure 1 the authors present calculated properties for five compounds.
    a) Were the molecules considered in their protonated form at the amino group?
    b) The compounds in panels A, B, and C do not have defined stereochemistry at the relevant carbon atom. The authors should specify the stereochemistry in each case. Alternatively, were the calculations performed using racemic mixtures?

  3. In the legend of Figure 3, please specify how the metabolism predictions were carried out.

  4. Why is “Scheme 1” not referred to as a “Figure”? The image appears blurry; the molecules should be redrawn using appropriate software such as ChemDraw. It seems here the authors want to do a self-citation.

  5. In Table 3, halogen bonds are described for LaSSBio-2129. However, this compound contains only fluorine in its structure, and fluorine cannot form halogen bonds—only hydrogen bonds. The authors should correct this and carefully review the entire table. Halogen bonds are a special type of interaction involving Cl, Br, or I with electronegative atoms such as O or N. Fluorine cannot undergo radial polarization forming a sigma-hole, and therefore cannot technically form halogen bonds, although it can participate in hydrogen bonding.

Author Response

For research article: Unraveling the Antihyperglycemic Effects of Dipeptyl Peptidase-4 Inhibitors in Rodents: A Multi-faceted Approach Combining Effect on Glucose Homeostasis, Molecular Docking, and ADMET Profiling

Response to Reviewer X Comments

In this work, the authors conducted a combined study evaluating the biological effects of DPP4 inhibitors. A large portion of the results are computational. Although the findings are not particularly striking when compared with existing compounds capable of inhibiting this enzyme, they do contribute information regarding subtle structural variations that could be useful for developing new and improved derivatives in the future.

Overall, the presentation is clear; however, while some general biological results are included, there are no selectivity studies of the proposed mechanism of action, nor are IC50 values reported for the studied compounds. This limits confidence in the proposed mechanism of action. I suggest the authors take a more critical view of their results and revise the work considering the following comments:

  1. The introduction would benefit greatly from a figure showing the chemical structures of DPP4 inhibitors (such as those discussed in the second paragraph) along with their IC50 values.

Response: We thank the Reviewer for this insightful comment regarding the inclusion of IC₅₀ values and selectivity data for the proposed DPP-4 inhibitors. We would like to clarify that such data have already been extensively determined and published in a companion article (Reina et al., RSC Advances, 2024, 14, 6617–6626. DOI: 10.1039/d4ra00450g). In that work, the series of β-amino-N-acylhydrazone derivatives (LASSBio-2123 to LASSBio-2130) were synthesized, fully characterized, and evaluated against human recombinant DPP-4. The reported IC₅₀ values ranged from low micromolar to sub-micromolar, with LASSBio-2127 showing IC₅₀ = 2.93 µM and the R-enantiomer LASSBio-2129 (6-R) exhibiting IC₅₀ = 5.08 µM, thereby validating their inhibitory potential relative to sitagliptin (IC₅₀ = 92 nM under the same experimental conditions) (Reina et al., et al 2024). Importantly, that publication also presented selectivity considerations, stereochemical effects, and preliminary structure–activity relationships. For clarity in the present manuscript, we have now cited this prior publication in the Introduction and Discussion sections, so that readers are directed to the experimental IC₅₀ and selectivity data already available in the literature. We believe this addition addresses the Reviewer’s concern while avoiding redundancy.

  1. In Figure 1 the authors present calculated properties for five compounds. a) Were the molecules considered in their protonated form at the amino group? b) The compounds in panels A, B, and C do not have defined stereochemistry at the relevant carbon atom. The authors should specify the stereochemistry in each case. Alternatively, were the calculations performed using racemic mixtures?

Response: Dear referee, thank you for your observation. Below is the detailed answer for alternatives a) and b):

  1. a) For the two-dimensional representation and the MLP map, the structures drawn in their neutral state were considered. However, for the lipophilicity calculation, the intrinsic value (logP) and the lipophilicity of the protonated structure in the amine (NH3+), called AZ logD in Table 1, were considered.
  2. b) For the structures cited by A, B, and C in Figure 1, the racemic mixtures of the compounds were considered, while for the structures cited by D and E, the discrepancy in results for both stereoisomers was considered.

  1. In the legend of Figure 3, please specify how the metabolism predictions were carried out.

Response: Dear referee, we appreciate your feedback. We have considered it and implemented it in the caption of Figure 3, adding technical and interpretative details on how the metabolism site prediction was made. We hope you will consider it for this new round of reviews.

  1. Why is “Scheme 1” not referred to as a “Figure”? The image appears blurry; the molecules should be redrawn using appropriate software such as ChemDraw. It seems here the authors want to do a self-citation.

Response: We thank the Reviewer for this valuable suggestion. We agree that the chemical structures should be presented with higher quality and in a clearer format. Following the Reviewer’s recommendation, we have redrawn the molecules using ChemDraw and replaced the original Scheme 1 with a high-resolution image, now referred to as Figure 12 in the revised manuscript.

Additionally, the caption was updated to clarify that the structures were adapted from Reina et al. (2024), thereby ensuring proper attribution while improving the readability of the manuscript.

  1. In Table 3, halogen bonds are described for LaSSBio-2129. However, this compound contains only fluorine in its structure, and fluorine cannot form halogen bonds—only hydrogen bonds. The authors should correct this and carefully review the entire table. Halogen bonds are a special type of interaction involving Cl, Br, or I with electronegative atoms such as O or N. Fluorine cannot undergo radial polarization forming a sigma-hole, and therefore cannot technically form halogen bonds, although it can participate in hydrogen bonding.

Response: Dear referee, thank you for your feedback. We apologize for this error. We have given your suggestion due consideration and implemented it in the molecular docking analyses, as well as in the figures and tables related to this result. We sincerely hope that this implementation meets your most valuable review criteria.

Round 2

Reviewer 1 Report

Comments and Suggestions for Authors

The authors have satisfactorily addressed my questions and implemented all requested corrections. The article can be accepted after the references are corrected. It is not necessary to resend the manuscript to me after this revision.

Reviewer 2 Report

Comments and Suggestions for Authors

The authors have addressed the most important comments, and in general, the manuscript is now better than the previous version. Therefore, It can now be considered for final publication.